# Deep Learning-Based Computer-Aided Fetal Echocardiography: Application to Heart Standard View Segmentation for Congenital Heart Defects Detection

**DOI:** 10.3390/s21238007

**Published:** 2021-11-30

**Authors:** Siti Nurmaini, Muhammad Naufal Rachmatullah, Ade Iriani Sapitri, Annisa Darmawahyuni, Bambang Tutuko, Firdaus Firdaus, Radiyati Umi Partan, Nuswil Bernolian

**Affiliations:** 1Intelligent System Research Group, Faculty of Computer Science, Universitas Sriwijaya, Palembang 30139, Indonesia; naufalrachmatullah@gmail.com (M.N.R.); adeirianisapitri13@gmail.com (A.I.S.); riset.annisadarmawahyuni@gmail.com (A.D.); bambangtutuko60@gmail.com (B.T.) virdauz@gmail.com (F.F.); 2Faculty of Medicine, Universitas Sriwijaya, Palembang 30139, Indonesia; radiyati.u.p@fk.unsri.ac.id; 3Division of Maternal-Fetal Medicine, Department of Obstetrics and Gynecology, Mohammad Hoesin General Hospital, Palembang 30126, Indonesia; nuswilbernoli@gmail.com

**Keywords:** fetal echocardiography, deep learning, fetal heart standard view, heart defect, instance segmentation

## Abstract

Accurate segmentation of fetal heart in echocardiography images is essential for detecting the structural abnormalities such as congenital heart defects (CHDs). Due to the wide variations attributed to different factors, such as maternal obesity, abdominal scars, amniotic fluid volume, and great vessel connections, this process is still a challenging problem. CHDs detection with expertise in general are substandard; the accuracy of measurements remains highly dependent on humans’ training, skills, and experience. To make such a process automatic, this study proposes deep learning-based computer-aided fetal heart echocardiography examinations with an instance segmentation approach, which inherently segments the four standard heart views and detects the defect simultaneously. We conducted several experiments with 1149 fetal heart images for predicting 24 objects, including four shapes of fetal heart standard views, 17 objects of heart-chambers in each view, and three cases of congenital heart defect. The result showed that the proposed model performed satisfactory performance for standard views segmentation, with a 79.97% intersection over union and 89.70% Dice coefficient similarity. It also performed well in the CHDs detection, with mean average precision around 98.30% for intra-patient variation and 82.42% for inter-patient variation. We believe that automatic segmentation and detection techniques could make an important contribution toward improving congenital heart disease diagnosis rates.

## 1. Introduction

Fetal echocardiography examination is widely applied in clinical settings due to its non-invasive nature, reduced cost, and real-time acquisition [1]. Such examination is usually assessed by ultrasound after an approximate gestational (menstrual) age of 18 weeks to find the heart structural abnormalities [2]. Assessment and evaluation of fetal heart abnormalities provide crucial information to families prior to the anticipated birth of their children about diagnosis, underlying etiology, and potential treatment options, which can greatly improve the survival rates of fetuses. One of the most common structural heart diseases is congenital heart defects (CHDs), which affect 5–9 out of 1000 births; CHDs cause 5% of all childhood deaths [2,3] and 18% of liveborn infants with CHDs die within the first year [4].

The process of CHDs examination begins with determining the location of the fetal heart based on four standard views, i.e., four-chamber (4CH), three-vessel and three-vessel trachea (3VV/3VT), and left and right ventricular outflow tract (LVOT/RVOT) [4]. The 4CH view is a basic standard fetal heart scan, whereas LVOT, RVOT, and 3VT views are complex fetal heart scans [3]. By using such views, the fetal heart anatomy abnormalities or CHDs can be detected. The previous result indicates that CHDs detection has improved from 55–65% in 4CH view evaluation only, and increased to 80–84% with combination view of LVOT, RVOT, and 3VT assessment [3,5]. However, physiological assessment to obtain fetal heart anatomy abnormalities utilizing such standard views requires well-trained and experienced maternal–fetal clinicians.

CHDs detection with expertise in general are substandard, the detection rates of only 30–50% [5]. Although a detailed quality control guideline has been developed to evaluate fetal heart standard planes, the accuracy of measurements remains highly dependent on humans’ training, skills, and experience [5]. Intra-observer and inter-observer variabilities exist in routine practice produce inconsistencies in image quality [6], which can lead to variances in the reading of specific heart anatomic structures [5,7]. In most cases of missed CHDs, either the fetal heart view is not correctly obtained, or the defect is clearly demonstrated but not recognized by the clinicians and operator [6]. Furthermore, there is a lack of well-trained clinicians in areas with poor medical conditions, making fetal echocardiography examinations impossible to perform. Previous work has shown a positive impact of increasing the operator experience and clinicians training programs on recognition of fetal heart anatomy [8,9,10,11,12]. Unfortunately, such programs are labor and time intensive and need to be repeated with staff turnover. To this end, automatic approaches to fetal echocardiography image quality assessment are needed to ensure that images are captured as required by guidelines and provide accurate and reproducible fetal heart biometric measurements.

Computer-aided diagnosis (CAD) with artificial intelligence (AI) can be used in fetal echocardiography image assessment to automatically segment and classify the fetal heart organ [11,12,13] and detect defects in the heart septum [7]. In the anatomically structure, CHDs condition commonly is recognized by a hole in atria, ventricle, or both, named atrial septal defect (ASD), ventricular septal defect (VSD), and atrioventricular septal defect (AVSD) [3,5], respectively. These conditions are very dangerous, as they allow shunt of blood flow from the right heart chambers to the left, and vice versa [3]. The deep learning (DL)-based convolutional neural networks (CNNs) architecture is an AI approach that can apply to fetal object diagnosis [6,7,9,10,11,12,13,14,15,16,17,18].

Several studies have reported powerful results regarding CNNs’ ability in segmentation, classification, and detection based on medical imaging [15,16,17,18,19,20,21]. CNNs, which are applications that perform adaptation functions without being specifically programmed, learn from data and make accurate predictions or decisions based on past data [6,7,8,9,10]. However, in the fetal echocardiography study based on CNNs leaks through missing boundaries caused by intra-chamber walls remain unresolved [11]. The previous studies proposed extracting heart structure patterns by selecting suitable regions of interest (RoIs) [10]. However, in the experiment, the explored object detection methods work for one candidate region only; they are hard to implement for detecting multiple candidates. This issue has been solved using a classification approach [20]. However, in this case, the CNNs is applied for only single task learning at a time, the process of segmentation, classification, or object detection is conducted separately in other words, these processes are not conducted simultaneously.

Multi-task learning in DL is essential for fetal heart imaging, as with the use of such a combination task, a model can segment multiple regions, select multiple candidates, classify multiple RoI, and detect multiple medical objects [11,12]. In this study, multi-task learning in terms of segmentation, classification, and detection processes are performed simultaneously for accurate fetal heart diagnosis. The contributions of this study are as follows:Propose a methodology for automatic segmentation with two-dimensional fetal heart echocardiography ultrasound images in normal and abnormal anatomic structure;Develops an instance segmentation approach for multi-task learning;Implements the proposed model by conducting the experiment with 24 objects, including 4 shapes of basic fetal standard views, 3 structural of congenital heart defect, and 17 objects of fetal heart chambers in 4 views;Validates the robustness of the proposed model with intra- and inter-patient scenario.

The remainder of this paper is organized as follows: Section 2 presents the details of the materials and methods. The experimental results and discussion are provided in Section 3. Finally, Section 4 concludes the study.

## 2. Materials and Methods

The general methodology of our study can be seen in Figure 1, the proposed workflow is divided into five main processes: data acquisition, data preparation, image annotation, deep learning model, and model evaluation. The workflow used in this study is utilized for automatic segmentation of the fetal heart standard view and heart defect detection. The whole process as summarized in the sub section.

### 2.1. Data Acquisition

In this study, the essential of anatomic structures used to evaluate image quality were defined by two senior maternal–fetal clinicians with experience in fetal echocardiographic examination at the Mohammad Hoesin Indonesian General Hospital. Four fetal heart standard views in normal anatomy were used, 4CH, 3VT, LVOT, and RVOT, whereas in the abnormal anatomy to assess CHDs, i.e., ASD, VSD, and AVSD conditions, only 4CH view was used. The four steps performed to prepare 2-dimension (2D) echocardiography image data are (i) data collection, (ii) ultrasound video conversion, and (iii) image cropping. The whole process of data acquisition is summarized as follows:The fetal echocardiography image was based on ultrasound video data collected from the Indonesian Hospital from 18–24 weeks pregnancy women in 4CH, 3VT, LVOT, and RVOT views with normal anatomy. Such video was recorded using a GE Voluson E6 with a loop length of 10 s to 5 min. An observational analytic study with a cross-sectional design was conducted to detect normal and abnormal anatomical structure fetal heart in utero. Due to the atrial and ventricular of the fetal heart are clearly visible in 4CH view, thus the cross-sectional fetal heart image for heart defects analysis only uses 4CH view.The ultrasound video was taken with several size variations of 1.02 megabytes to 331 kilobytes. All ultrasound videos should be transformed into frames and then resized to a resolution of 400 × 300 pixels. All fetal heart images were retrieved for retrospective analysis using the digital imaging and communications in medicine (DICOM) format. The framing process from video to 2D images utilize the cv2.VideoCapture() function. The ultrasound video was read frame by frame into the new size, and all the generate frames were stored in frame storage using the cv2.imwrite() code to create ground truth images.The whole fetal heart images generated by software are verified by maternal–fetal clinicians in the department of obstetrics and gynecology, General Hospital Mohammad Hoesin Palembang, Indonesia. By using the cropping process, unnecessary information of the raw images was removed. The outputs were coded after cropping from the echocardiogram video using output_movie.release(). All processes were run on the Python OpenCV library.

### 2.2. Data Preparation

The fetal heart ultrasound datasets collection comprised four views of imaging planes of normal and defective fetal hearts. All images were labeled in accordance with widely used fetal heart anatomical planes by a maternal–fetal clinician. The dataset represented a real clinical setting and the ultrasound video data were acquired during standard clinical practice in one year (between 2020 and 2021). However, due to the pandemic situation, only about 100 pregnant women attending for routinary pregnancy screening during their second and third trimesters (18–24 weeks) were included in this study. From the whole data, the CHDs condition is hard to find, therefore only 20 pregnant women included in this study exhibited abnormal anatomy, alongside 30 pregnant women with normal anatomy.

Each ultrasound video for one patient produces 40 images; thus, 50 pregnant women produce about 2000 images in normal and abnormal anatomy. The maternal–fetal clinician selected images belonging to the four anatomical planes most widely used in routine maternal–fetal screening. The clinician selected only images complying with the minimum quality requirements, only a clear cross-sectional scan image was included to process further. It consists of 332 images for fetal standard view segmentation and 917 images for heart defect detection (refer to Table 1). The training process randomly split the collected cases into a training set and a validation set, and the model established by the training set data was tested against the validation set in order to ensure the accuracy and stability of the model.

Images with inappropriate anatomical planes (cropped or badly captured) and those with calipers were excluded. The dataset composition was clearly imbalanced (some classes were more frequent than others), as is usually the case in real clinical scenarios. The sample of the raw ultrasound image was based on four views in normal anatomy, as depicted in Figure 2. In such sample, there are the left atrium (LA), left ventricle (LV), right atrium (RA), right ventricle (RV), ductus arteriosus (DUCT), superior vena cava (SVC), aorta ascendens (AoA), aorta descendens (Ao), and main pulmonary artery (MPA), whereas the sample of the raw ultrasound image of abnormal anatomy structure, with the three heart defects such as ASD, VSD, and AVSD condition, is compared to normal anatomy structure in Figure 3. In the abnormal structure, there are hole (H) as heart defect in each condition. Each defect has the variation of hole size; such hole size indicates the disease severity. However, in this study, we only detected the hole, without measuring the hole size.

### 2.3. Image Annotation

The anatomical heart structures are critical for the segmentation process. The maternal–fetal clinician as the image annotator should drew precise boundaries around the heart images manually with data annotation tool (LabelMe) [7]. LabelMe to provide an online annotation tool to build image databases for computer vision research. The significant variations in image quality, shapes, sizes, and orientations between the pregnant women were used to create a database of ground truths. In the fetal echocardiography with normal anatomy, each standard view has a different structure of heart chamber; therefore, the annotation should be conduct for all standard views with their respective chamber such as, 4CH standard view consists of five heart chambers, i.e., Ao, LA, LV, RA, and RV; 3VT standard view consists of three heart chambers, i.e., DUCT, SVC, and AoA; LVOT standard view consists of five heart chambers, i.e., AoA, LA, LV, RA, and RV; and RVOT standard view consists of four heart chambers, i.e., DUCT, SVC, AoA, and MPA.

Especially for heart defect detection, only 4CH view was used to analyze ASD, VSD, and AVSD images. Annotated images indicate the position of defect in the atrium, ventricle, or both of them. Figure 4 depicts the sample of annotated images for a standard view of 4CH, 3VT, LVOT, and RVOT, and Figure 5 shows the sample annotated images of defect position in ASD, VSD, and AVSD. Finally, the whole annotated images are labelled as the ground truth database, and it was saved in the JSON file format (json).

### 2.4. Deep Learning Model

In this study, the instance segmentation approach is developed based on Mask-RCNN architecture (refer to Figure 6) [18,22]. The Mask-RCNN structure has two main processes, region proposal networks (RPNs) as feature extraction and fully convolutional networks (FCNs) as multi-task learning process in terms of simultaneous classification, detection, and segmentation.

#### 2.4.1. Region Proposal Networks

The input of the region proposal networks (RPNs) is 2D ultrasound images; all the fetal hearts have the same size of resolution around 400 × 300 pixels. The ResNet50 architecture was applied as the backbone in the RPNs for the feature extraction mechanism. It can represent more complex functions and learn features from different network levels, from edges (shallower layers) to very complex features (deeper layers). The RPNs use to generate RoIs, which will be used to predict classes and generate masks. Each RPNs had five convolutional layers, which were used to process high-level feature inputs to low-level outputs. The ResNet 50 structure as seen in Table 2, and the example feature map from ResNet 50 as seen in Figure 7.

Mask-RCNN adds learning process for segmentation masks in each RoI. The segmentation process is simultaneous with each of the other processes (bounding box regression and class generation), using convolution arrays (feature map) from RPNs. The RPNs classify the feature and tighten bounding boxes with region of interest (RoI) alignment (RoIAlign). The hyperparameters used in the RPNs’ structure are batch size for each image of 256; learning rate of 0.001; momentum of 0.9; non-maximum suppression threshold of about 0.7; intersection over union (IoU) baseline of 0.5; and anchor sizes of 32, 64, 128, 256, and 512. RoI alignment was performed to pool all RoIs remaining on the feature maps to a fixed size. As the regression model produced the RoI position, it was generally a floating-point number; however, the pooled feature map required a fixed size.

#### 2.4.2. Fully Convolutional Networks

Fixed-size RoIs were sent to the fully convolutional networks (FCNs) for object classification, detection, and segmentation. The mask branch module is a small FCN applied to each RoI, and it predicts a segmentation mask for each pixel. In this study, fetal heart area segmentation was a parallel branch to the wall-chamber classification and bounding box regression of the heart position. The FCNs utilize stride 2 and 3 × 3 max pooling, with the Softmax as the objective function. The general FCNs structure use in fetal heart segmentation as seen in Table 3.

#### 2.4.3. Training and Validation Performance

By using Mask-RCNN approach, the system can recognize the objects’ classes, locations (the bounding box), and shapes [22]. The proposed model utilizes a multi-task learning with the loss function that incorporates losses from predictions in classification, detection, and segmentation for each instance [22]. The first term in the loss function is $L_{\mathrm{cls}}$, which measures the error in the predicted class label. The class prediction branch uses a SoftMax layer to output the final class predictions for each instance. For instance, i, the class prediction, is a vector denoted by $p^{\to i}$. Each element $p_{j}^{\to i}$ exists in the interval (0, 1) and is interpreted as the predicted probability that instance i belongs to class j. If the true class of an instance i is u, then $L_{\mathrm{cls}}$ is given by the log loss function (Equation (1)).
(1)$$L_{\mathrm{cls}}{(p}^{\to i}{,u)=-\mathrm{logp}}_{u}^{\to i}$$

The second term is $L_{\mathrm{bbox}}$, which measures the error predicted bounding boxes. The ground truth for the bounding box for an instance of class u is given by the vector $\vec{v}=(v_{x},v_{y},v_{w},v_{h})$, where the four indices indicate the x and y coordinates of the center of the box, w width of the box, and h height of the box. Detailed information about the format of the bounding boxes is given in [22]. The predicted bounding box is denoted by $\vec{t}$ and has the same form as $\vec{v}$. $L_{\mathrm{bbox}}$ is given by following equation:(2)$$L_{\mathrm{bbox}}=\underset{i\in x,y,w,h}{\sum }{\mathrm{smooth}}_{L1}{(t}_{i}^{u}{-v}_{i})$$
(3)$${\mathrm{smooth}}_{L1}\left(x\right)=\left\{\begin{array}{c} 0{.5x}^{2}\\ \left|x\right|-0.5 \end{array}\right.\;\begin{array}{c} ,\;\mathrm{if}\;\left|x\right|\text{<}1\\ ,\mathrm{else} \end{array}$$

The third term is $L_{\mathrm{mask}}$, which measures the error predicted segmentation masks for each instance. In the mask prediction branch, a sigmoid activation is applied to every pixel in the final feature map. The sigmoid value bounds at 0 to 1 and is interpreted as the probability that a given pixel is included in the proposed segmentation mask. Then, $L_{\mathrm{mask}}$is given by the binary cross-entropy between the predicted and ground truth masks. Let $Y_{i}$ and ${\overset{⌢}{p}}_{i}$ correspond to the ground truth pixel label (0 or 1) and the predicted probabilities for pixel i, respectively. For ground truth and predicted masks with *N* total pixel $L_{\mathrm{mask}}$ is presented in Equation (4).
(4)$$L_{\mathrm{mask}}=-\frac{1}{N}\overset{N}{\underset{i=1}{\sum }}Y_{i}\log\overset{◠}{p}{i+(1-Y}_{i})\log(1-\overset{◠}{p}i)$$

If the prediction is given by the categorical cross-entropy, then SoftMax activation function is applied, and $L_{\mathrm{mask}}$ is presented as Equations (5) and (6) denotes the total loss of the model.
(5)$$L_{\mathrm{mask}}=-\frac{1}{N}\overset{{\sum^{◠}}_{i}}{\underset{i=1}{\sum }}\log$$
(6)$${L=L}_{\mathrm{cls}}{+L}_{\mathrm{bbox}}{+L}_{\mathrm{mask}}$$

### 2.5. Model Evaluation

In order to validate and evaluate the performance of the instance segmentation model, the outputs of Mask-RCNN are validated by using six metrics, i.e., loss in classification, loss in segmentation, loss in detection, overlapping between the annotated and predicted inputs of each class in IoU, Jaccard index or Dice coefficient similarity (DCS) for segmentation, and mean average precision (mAP) for object detection [7,17,19].

The DCS is a Jaccard similarity coefficient used for gauging the similarity and diversity of sample sets. In this case, to measure the performance of predictive images with detailed truth labels. The DCS is illustrated in Equation (7).
(7)$$\mathrm{DCS}\;\left(X,Y\right)=\frac{2\sum_{i}^{N}X_{i}Y_{i}}{\sum_{i}^{N}X_{i}^{2}+\sum_{i}^{N}Y_{i}^{2}}$$
where N is the number of runs of the predicted results, $X_{i}$ is the prediction result, and $y_{i}$ is the truth label. The pixel index value of the DCS, which is in the interval [0, 1], measures the match probability between the predicted and ground truth images.

The performance of the trained/validated Mask R-CNN model was quantitatively evaluated by mAP. The mAP score is a widely adopted metric for assessing object detection models. The mAP values of the various groups were computed, and their average was obtained. Although the model would detect various objects, the classes to be assigned to these objects were not always certain. However, even if the expected class for an object or instance were correct, the output criterion must still look at how well the model locates it spatially in the picture. Equation (8) depicts the commonly used mAP.
(8)$$\mathrm{mAP}=\frac{1}{n_{\mathrm{cl}}}\frac{\sum_{i}n_{\mathrm{ii}}}{t_{i}}$$
where $n_{\mathrm{cl}}$ is the total of all the different classes and $t_{i}=\sum_{j}^{}n_{\mathrm{ij}}$ is the total number of pixels of class i.

## 3. Experimental Results and Discussion

In ultrasound examination, the position of fetal heart is difficult to predict due to small size and unpredictable shape and orientation. Fetal–maternal clinicians conduct examinations to determine a fetus’ condition in the womb (whether it has a congenital heart defect) before birth. In this study, we propose the comprehensive computer-assisted echocardiographic interpretation is determining whether computers can learn to recognize such condition. To ensure the performance of the learning process, all the networks are trained in the computer specifications as follows: the processor was an Intel^®^ Core™ i9-9920X CPU @ 3.50GHz and 490191 MB RAM, the GPU was a GeForce 2080 RTX Ti, by NVIDIA Corporation GV102 (rev a1); the operating system was Windows 10 Pro 64-bit (10.0, Build 18363).

### 3.1. Fetal Heart Standard View Segmentation

We benchmarked widely used state-of-the-art CNNs-based Mask-RCNNs with three different backbone architectures: ResNet50, ResNet101, and MobileNetV1. The networks’ original architecture of Mask-RCNN was maintained in all cases. All networks were first pre-trained using the Microsoft common objects in context (COCO) dataset, then fully retrained using our training data to produce the probability scores for each class. We conduct the fetal heart standard view segmentation with normal anatomy of the 4CH view, the expected normal appearance of the LVOT/RVOT view and the additional views required for the complex ultrasound obstetric images with 3VT view. Whereas fetal heart abnormality anatomy examination by using only the 4CH view.

The performance of Mask-RCNN in fetal heart standard view segmentation can be seen in Table 2, which shows that ResNet50 outperformed ResNet101 and MobileNetV1 in terms of the mAP, IoU, and DCS. ResNet50 produced average mAP, IoU, and DCS values of 96.59%, 79.97%, and 89.70%, respectively. All values exceeded 50%, given that the baseline of IoU was 50%, and those of mAP and DCS were over 70%. Therefore, the Mask-RCNN model with the ResNet50 architecture could detect all heart chamber in the four views.

Table 4 shows the performance of fetal heart standard view. The experimental result showed that the heart chambers in the LVOT view were the most difficult to detect based on three architectures. There were several ambiguities between the 4CH and LVOT cases, as the appearance of the fetal heart is similar between these views. 4CH has four chambers, whereas LVOT has five chambers with ascending aorta. However, ascending aorta looks faint, as it is close to the valves in the fetal heart. It is differentiated only by subtle, indistinct structures, such as heart valves, which varied significantly in the ultrasound image artefacts and the relative movement between probe and fetus. It is a long-axis view of the heart, highlighting the path from the left ventricle to the ascending aorta with five-part of the heart chamber. The detection result produced a 60% IoU, but the DCS value reached 86.55%.

Generally, maternal–fetal clinicians use their judgement to determine whether certain heart substructures are in the correct anatomical localizations by comparing normal and abnormal fetal heart images. Four standard views in the ultrasound images are used in examinations to perform fetal heart diagnoses. In this study, the fetal heart view was segmented automatically using the proposed model. Fetal heart chamber as an object should be detected and segmented in the four fetal heart standard views namely, AoA, AoD, LA, RA, LV, RV, DUCT, SVC, and MPA. Figure 8 and Figure 9 presented the heart chamber prediction performance for a standard fetal heart scan in terms of the IoU and DCS (Jaccard index) performance. A total of 17 heart chambers are needed to be segmented and detected: five objects for 4CH view, three objects for 3VT view, five objects for LVOT view, and four objects for RVOT view.

The IoU and DCS performance shows that the instance segmentation with the ResNet50 architecture as the backbone produced excellent predictions for all chambers in each view. Therefore, the Mask-RCNN with the ResNet50 architecture as the backbone of RPNs could segmented and detected the object based on the annotated RoI. In Figure 10a, the sample of segmentation result of fetal standard heart view is provided, and Figure 10b shows the heart chamber segmentation is presented separately. The standard view segmentation, to mark the shape of the cross sectional of the fetal heart, and the heart chambers segmentation, to show the part of each cross-sectional, belong here, whereas in Figure 11a–d, we experimented on two combinations, in such process a fetal heart view and heart chamber is merged, with about 17 heart chamber objects and four heart standard views to predict. Figure 11a,d shows the sample of segmentation results with different colors, but each object has the same description as Figure 10a,b. Based on the proposed model, all objects can be predicted with satisfactory performance (about 96.59% mAP, 79.97% IoU, and 89.70% DCS). The high mAP shows that the object detection process based on the proposed model obtained the overlapping area between the annotated and predicted RoIs of each bounding box close to 100%. The proposed Mask-RCNN model with ResNet50 yielded a 3.41% error in prediction between the annotated and predicted RoIs.

### 3.2. Heart Defect Segmentation and Detection in 4CH View

The fetal heart anatomy in 4CH view showed the expected normal appearance [23]. As apical 4CH is the original gold standard view in fetal echocardiography, inability to image this should alert the scanner about a potential problem [24]. This view should not be mistaken for a simple chamber count as it involves a careful evaluation of specific criteria [24]. Based on such criteria, the detection of the fetal heart abnormality was screened only by 4CH view [8]. Three CHD conditions (with defects in atria, ventricles, and both) were measured with the IoU and DCS values. The minimum IoU value for detecting the defect object in each fetal heart was 0.5. High IoU and DCS values indicated that the defect prediction overlapped with the proposed architectural model, which is almost similar to the ground truth.

Two scenarios for the learning processes were conducted in this study based on intra- and inter-patient variation data. Intra-patient variations meant that a fetal heart image coming from the same patient was split for the testing process. Inter-patient variations meant that the tested fetal heart images were from different patients. In the intra-patient data for ASD, VSD, and AVSD, the proposed model produced IoU and DCS values exceeding 50%. However, for the inter-patient data, although a 55.99% IoU was obtained for ASD, the IoU values of VSD and AVSD were under 50%. The DCS value exceeded 50% for ASD and AVSD, but that of VSD was only close to 50% (refer to Table 5).

Overall, the defect detection performance reached over 50% in IoU and DCS for intra-patient data. The inter-patient data were hard to detect due to large variations in fetal heart images, size of defect, and image quality, especially in VSD and AVSD condition. The result was under 50% IoU; all measurement decreased about 13 to 15% if the proposed model was tested with unseen images.

The sample image of heart defect segmentation and detection is depicted in Figure 12. In the 18–21 weeks of pregnancy, the fetal heart has size around 24 mm [23], thus the hole (defect) size in the heart septum will have a size <24 mm. At this stage of development, therefore, it remains difficult to visualize with precision the details of cardiac anatomy as seen during fetal echocardiography. By using our proposed model, it can be segmented and detected with IoU and DCS about 59% and 69%, respectively, in intra-patient, and about 47% IoU and 57% DCS in inter-patient scenarios. This means our model has the ability to segment and detect until a 50% overlap with the ground truth.

This study determined the mAP value for each defect condition (ASD, VSD, and AVSD) in addition to the IoU and DCS values. A high mAP value indicated that the defect prediction from the model was similar to the ground truth generated by the maternal–fetal consultant. Table 6 shows the object detection results with mAP performance; the highest mAP value (98.30%) was obtained from the intra-patient data; however, the mAP decreased to 82.42% in the inter-patient data. CNN-based instance segmentation works using a simple linear iterative clustering algorithm, which takes an image as input and outputs its division into super-pixels. The proposed model measures the overlap between the annotated input and predicted target, but it does not label all the image pixels, as it segments only the RoI. Therefore, if the input image is new (from inter-patient), the detection performance will decrease, but its performance still satisfactory due to the reduction only 16% with the mAP value over 80%.

We also conducted heart chamber segmentation and detection in 4CH view with abnormal anatomy image. The fetal heart chamber prediction with the proposed model is presented in Figure 13. This experimental result differed from the IoU and DCS performance in Figure 11, as the fetal heart images are taken from the patients with CHDs. The experiment was conducted based on intra- and inter-patient data. The RoI was a segment of four object classes, namely, LA, LV, RA, and RV. With the use of the Mask-RCNN model, all classes can be segmented and classified in the three conditions. Overall performances show that intra-patient data allowed better IoU and DCS performance compared with inter-patient data.

As shown in Figure 13a,d, the proposed model produced satisfactory results, with a large overlap between the ground truth and the predicted image. All IoU values exceeded the baseline of 0.5, which is the gold standard value for ensuring that all processes can be run with good performance. The IoU and DCS performances with the intra-patient data were better than those with the inter-patient data, with scores of above 66.37% and 79.60%, respectively. The performance with the inter-patient data was poorer than that with the intra-patient data. Due to the inherent differences in appearance across different imaging modalities, it is challenging to construct accurate image similarity measures. As the underlying anatomical components vary between patients, inter-patient registration might be difficult. In future work, the detection performance for the inter-patient scenario should be enhanced. The image sample for the heart chamber with a heart defect can be seen in Figure 14 with the defect position marked in red and blue. In this detection process, the defect can be small or large, depending on its severity. However, in this study, the defect size parameter was not taken into account; for further research, it will be very important to diagnose the severity of the condition.

The proposed instance segmentation model in RPNs and FCNs block can simultaneously perform three processes: classification, detection, and segmentation. Therefore, three losses can be achieved, namely, object detection loss or bbox loss, classification or class loss, and segmentation or mask loss. During the training phase, we added an early stopping mechanism in order to prevent the model becoming overfit to the training data. We monitor the change of validation loss in order to stop the training. As result, the loss curve can be seen in Figure 15, all the response in RPNs and FCNs in the training and validation processes decreased to the stability (zero) point; the gap between the two curves of training/validation was relatively small. The RPNs’ response reaches around 0.1 to 0.25 detection loss in training and validation, and around 0.003 classification loss in training and validation. In the FCNs’ response, produce detection loss in training and validation was around 0.05 to 0.12, classification loss in training and validation was around 0.02, segmentation loss in training and validation was around 0.07 to 0.1, and finally total loss in the proposed model in training and validation was around 0.3 to 0.8. Hence, we concluded that the proposed model did not experience overfitting during the training process, despite the limitations in the training data.

### 3.3. Benchmarking Our Model with Existing Studies

For benchmarking, we compared the proposed model with those from other authors in medical cases, as presented in Table 7. For making a fair comparison, all selected methods are based on the segmentation and object detection approach with Mask-RCNN architecture, and the mAP was used. The mAP is a metric to measure the sensitivity of the model, the high of mAP performance indicates a model that is stable and consistent across difference confidence threshold.

Table 7 provides several existing segmentation studies for psoriasis skin [19], endoscopy disease [17], exudates and microaneurysms [25], brain tumor [26,27], lung nodule [28], breast tumor [29], and nuclei [30]. Mask-RCNNs for medical imaging utilize the instance segmentation models use super-pixels as the base for segmentation process, as a graphical preparatory clustering method. It clusters pixels in the vicinity in geometric and color spaces prior to object segmentation using a simple linear iterative clustering algorithm [17,18]. However, it does not label all of the image pixels, as it segments only the RoIs. From previous studies [17,25,31], the segmentation rate was unsatisfactory, producing mAP values around 0.5. This happens as RGB images differ with a large pixel variation; thus, they cannot follow a distribution in [17]. In Shenavarmasouleh et al. [25], exudates and microaneurysms segmentation has minimum prediction confidence hyperparameter of 0.35 as standard threshold (normally about 0.50), whereas a completely correct prediction will result in 1.0. In Vuola et al. [30] nuclei have a variety of cells acquired under various conditions produce the shift in different RGB datasets are significantly large. To improve the performance, they ensemble Mask-RCNN with U-Net architecture, however more pixel-wise processes are involved, which increases the time consumption and computation cost. In addition, the masks from the models do not exactly fit the object image, and not every image pixel is marked separately. From all studies, the masks that they had from the datasets were only associated with one type of object and, for the most part, minimum overlap between the two datasets.

A different study produced a mAP of over 0.75; the super pixels of each object image were similar between the ground truth and the prediction for the pixel-level instance performed the same process [19,26,27,28,29,31]. However, in [19,26], they added data augmentation and other preprocessing techniques; data augmentation arises from the data bias, as the augmented data distribution can be quite different from the original one. This data bias leads to a suboptimal performance. Study [27] produced satisfactory results with higher mAP with only two classes, tumor and non-tumor. Similarly, with other studies in [28,29,31] the instance segmentation approach can segment the object with best mAP performance. However, they only use two classes, healthy and non-healthy lesion, whereas the instance segmentation is prepared for multi-classes and multi object segmentation.

Our proposed model, the Mask-RCNN based on ResNet 50 backbone, performed well with intra- and inter-patient data for two objects fetal heart views and fetal hear defects. We conducted the experiment with multi-object and multi-class segmentation for 24 medical objects. The heterogeneity of data types from the various modalities and clinical challenges caused by variations in the local textures was not an obstacle to produce satisfactory performance in identifying pathologies about 0.98 mAP. The RoI regions were automatically delineated, and all features were extracted from raw images by CNNs with ResNet 50 architecture, layer by layer, without previously giving the features. As a result, the proposed method has the advantage of observation anatomical structure comprehensively, not only by analyzing single features. This means the proposed model can segment four fetal heart views and has the ability to classify the heart chamber and aorta in each view, also detecting the hole as a defect in heart septum.

To our knowledge, no study has been conducted on fetal heart view segmentation and heart defect detection using an instance segmentation technique. Although the results are promising, this study has some limitations, (i) only fetal heart on 4CH view was used in this study for the CHDs detection case, (ii) the defect size was not taken into account to diagnose the CHDs severity, and (iii) the number of fetal heart image data population with normal and abnormal fetal heart structure should be added, in order to increase the inter-patient performance result. Furthermore, many other methods could have been benchmarked, and computational models could have benefited from the application of previous steps, such as image segmentation, instead of analyzing images as a whole.

## 4. Conclusions

Deep learning is a data-hungry method, but we showed a surprisingly small number of fetal echocardiography images can be used to significantly boost diagnosis from what is commonly found in practice. We conducted the experiment by selecting the input data according to clinical recommendations for only four fetal heart standard views rather than the entire ultrasound. This strategy allowed us to reduce the size of the input data to our diagnostic model and thereby achieve computational efficiency. Due to this, efficiency in prediction is key to translating this study toward real-world, resource-poor settings. Quantitative measures of fetal structure and function approximated clinical metrics and followed patterns found in normal and abnormal structure. A straightforward model, ours is an effective method of segmenting four fetal heart views, classifying the heart chamber and aorta in each view, and detecting the hole as a defect in heart septum in intra-inter-patient scenario. As a result, our proposed model achieved a remarkable performance, with 98.3% and 82.42% in intra-patient and inter-patient scenarios, respectively. We look forward to testing and refining these models in larger populations in an effort to democratize the expertise of fetal cardiology experts to providers and patients worldwide.

## Figures and Tables

**Figure 1 sensors-21-08007-f001:**
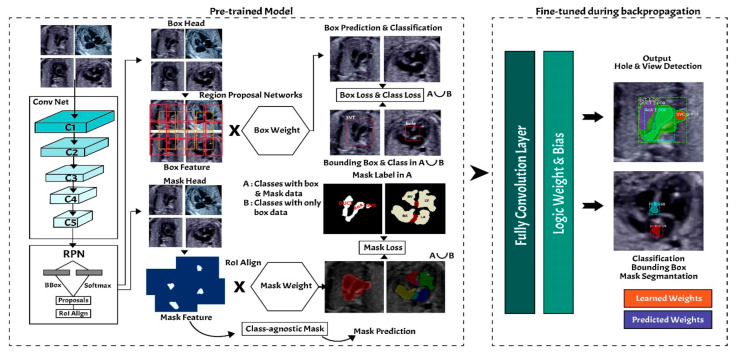
The proposes workflow of fetal heart standard view segmentation for heart defect detection with instance segmentation approach.

**Figure 2 sensors-21-08007-f002:**
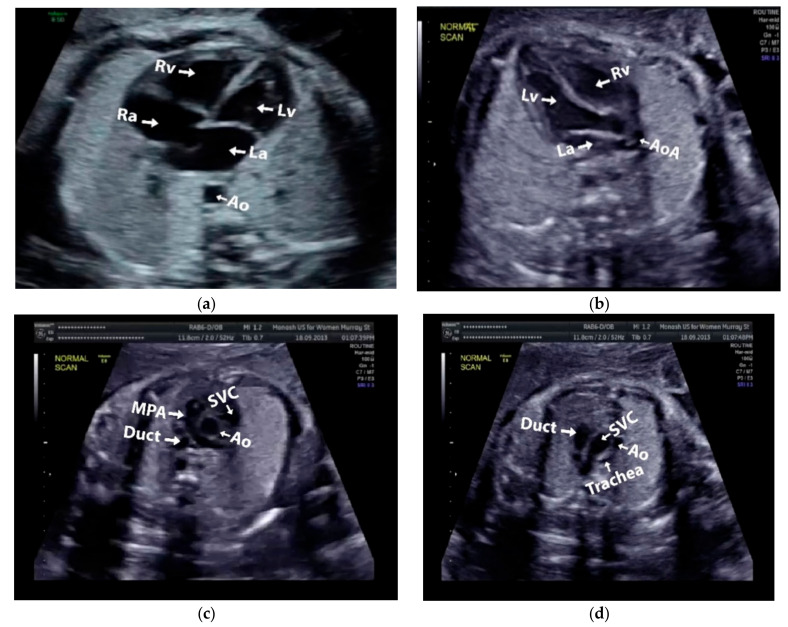
Fetal heart scan in four standard views of normal anatomy: (**a**) 4CH; (**b**) LVOT; (**c**) RVOT; and (**d**) 3VT.

**Figure 3 sensors-21-08007-f003:**
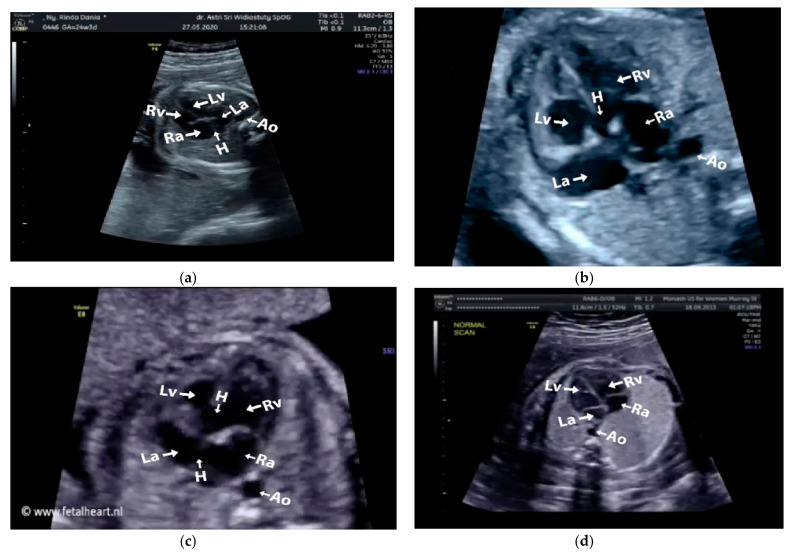
Fetal heart scan in 4CH view for CHDs detection: (**a**) ASD; (**b**) VSD; (**c**) AVSD; and (**d**) Normal.

**Figure 4 sensors-21-08007-f004:**
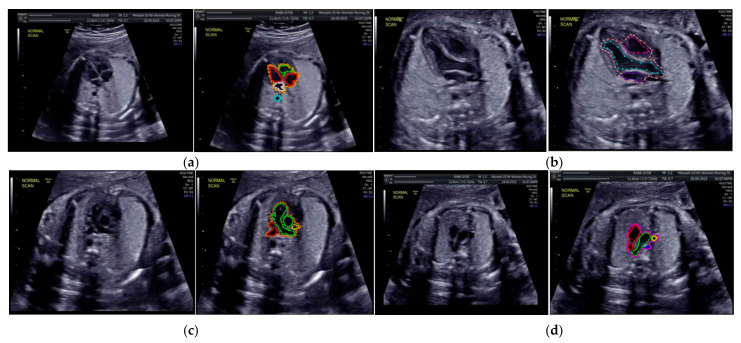
The sample of annotated images by maternal–fetal clinician for standard fetal heart view segmentation in (**a**) 4CH (orange: view, cyan: AoA, red: LA, grey: RA, green: LV, and red: RV); (**b**) LVOT (orange: view, cyan: LA, purple: RV, and blue: LV); (**c**) RVOT (orange: view, green: MPA, red: DUCT, and yellow: SVC); and (**d**) 3VT (purple: view, yellow: AoA, green: SVC, and red: DUCT); based on normal anatomy.

**Figure 5 sensors-21-08007-f005:**
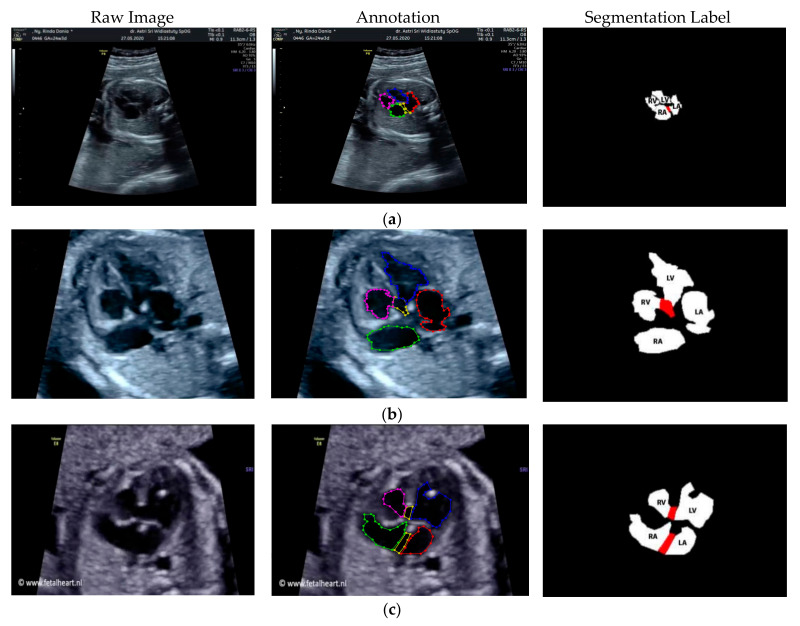
The sample of annotated image by maternal–fetal clinician for heart defect detection in case: (**a**) ASD; (**b**) VSD; and (**c**) AVSD. In the annotation, the green line is RA, the red line is LA, the purple line is RV, the blue line is LV, and the yellow line is defect.

**Figure 6 sensors-21-08007-f006:**
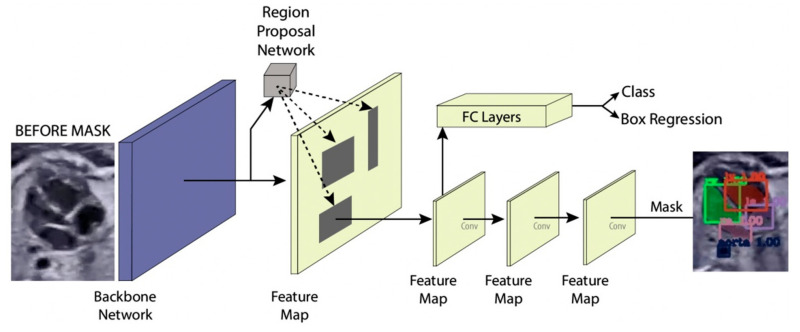
Instance segmentation approach.

**Figure 7 sensors-21-08007-f007:**
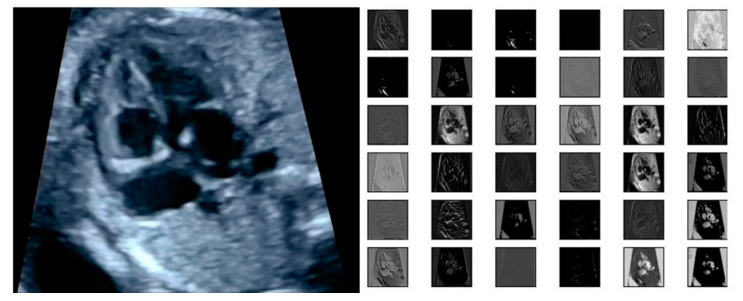
The example of feature map extracted from ResNet50 architecture in the RPNs back bone.

**Figure 8 sensors-21-08007-f008:**
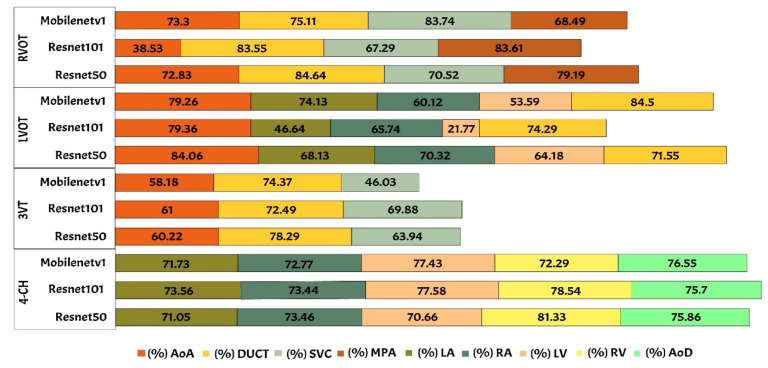
The IoU performance in heart chamber segmentation in four fetal heart standard views.

**Figure 9 sensors-21-08007-f009:**
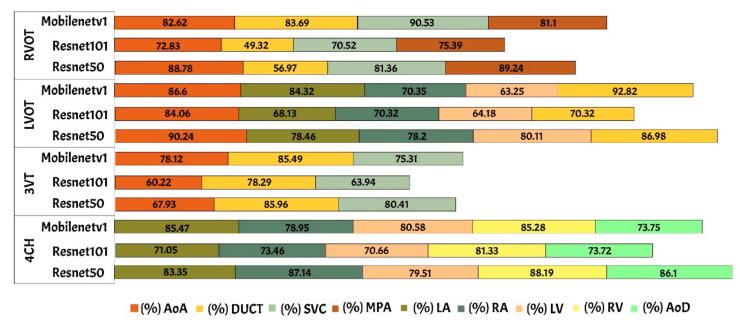
The DCS performance in heart chamber segmentation in four fetal heart standard views.

**Figure 10 sensors-21-08007-f010:**
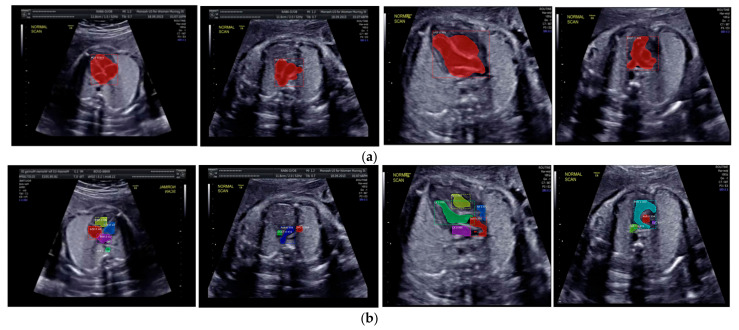
The sample segmentation result of standard view and heart chamber for normal heart anatomy structure: (**a**) red color contour denotes the fetal heart boundary segmentation in each view, from left to right are 4CH, 3VT, LVOT, and RVOT; (**b**) heart chamber segmentation in each view from left to right are 4CH (red: RA, purple: LA, yellow: RV, and blue: LV), 3VT (green: DUCT, blue: AoA, and red: SVC), LVOT (green: LV, red: AoA, blue: RA, and yellow: RV), and RVOT (green: DUCT, cyan: MPA, red: AoA, and purple: SVC).

**Figure 11 sensors-21-08007-f011:**
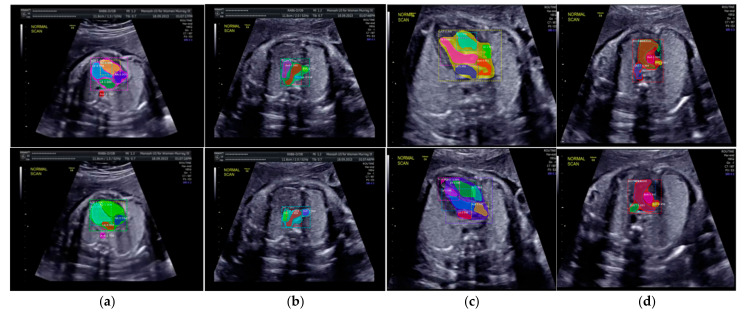
Fetal heart view with heart chamber segmentation in (**a**) 4CH, (**b**) 3VT, (**c**) LVOT, and (**d**) RVOT for normal heart anatomy structure. Fetal heart view boundary and heart chamber part as the same description with Figure 10.

**Figure 12 sensors-21-08007-f012:**
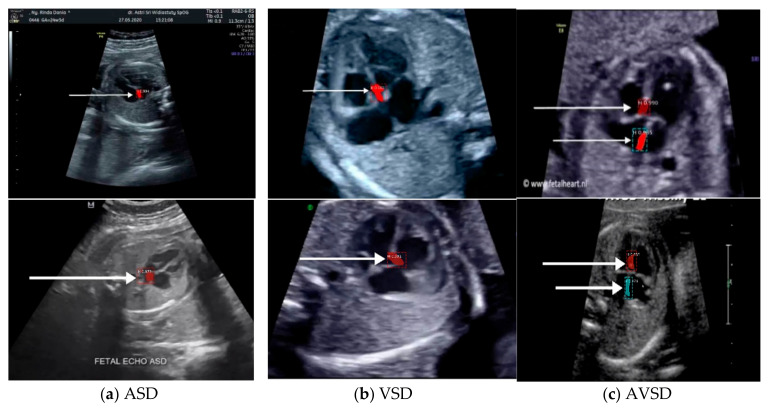
Sample image result of CHDs detection with 4CH view. The white arrow indicates the defect, whereas red and blue colors are the defect position in the heart septum.

**Figure 13 sensors-21-08007-f013:**
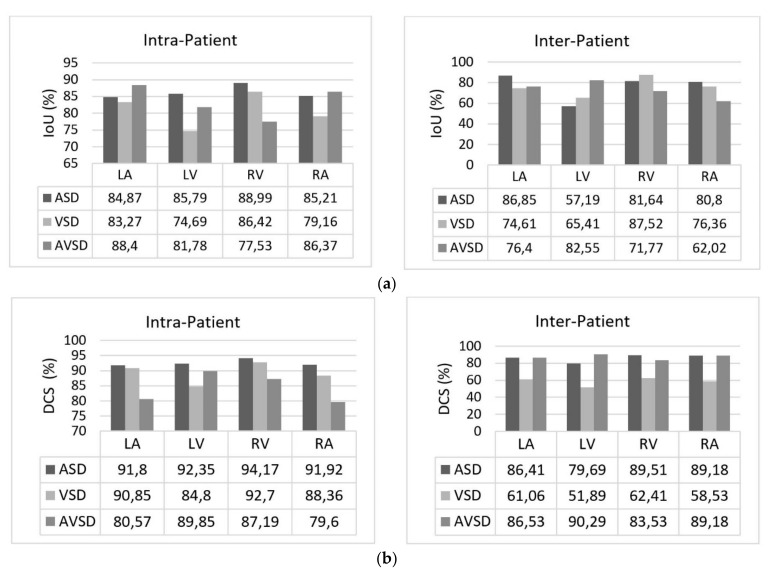
The performance in fetal heart chamber segmentation in 4CH view based on intra- and inter-patient scenario: (**a**) IoU and (**b**) DCS.

**Figure 14 sensors-21-08007-f014:**
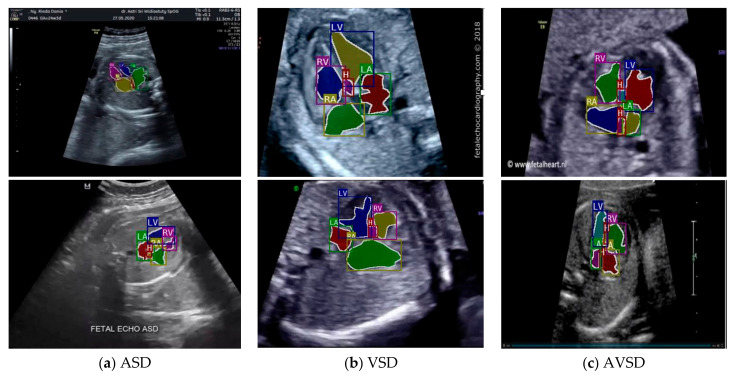
The sample result of wall-chamber segmentation with 4 CH view in ASD, VSD, and AVSD condition based on abnormal anatomy structure.

**Figure 15 sensors-21-08007-f015:**
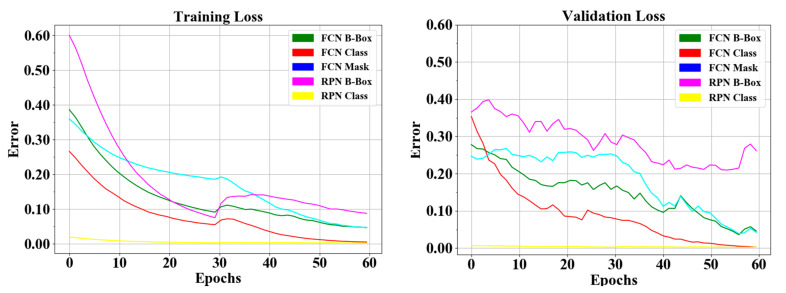
Loss curve of heart defect detection with proposed instance segmentation model. RPNs and FCN loss in training and validation set.

**Table 1 sensors-21-08007-t001:** Data distribution of fetal heart view and CHDs images. All data are extracted from ultrasound video from normal and abnormal anatomy.

Class	Training	Validation	Testing	Total
3VT view	58	8	6	72
4CV view	75	8	11	94
LVOT view	29	4	3	36
RVOT view	23	4	3	30
ASD	200	9	24	233
VSD	115	10	16	141
AVSD	168	9	18	195
Normal	303	10	35	348
Total				1149

**Table 2 sensors-21-08007-t002:** ResNet50 Structure.

Layer Name	Output Size	Output Shape
Conv 1	112 × 112	7 × 7, 64 Stride 2
Conv 2	56 × 56	3 × 3 max pool, stride 2
		$\left[\begin{array}{c} 1\times 1;\;64\\ 3\times 3;\;64\\ 1\times 1;\;256 \end{array}\right]\times 3$
Conv 3	28 × 28	$\left[\begin{array}{c} 1\times 1;\;128\\ 3\times 3;\;128\\ 1\times 1;\;512 \end{array}\right]\times 4$
Conv 4	14 × 14	$\left[\begin{array}{c} 1\times 1;\;256\\ 3\times 3;\;256\\ 1\times 1;\;1024 \end{array}\right]\times 6$
Conv 5	7 × 7	$\left[\begin{array}{c} 1\times 1;\;512\\ 3\times 3;\;512\\ 1\times 1;\;2048 \end{array}\right]\times 3$
-	1 × 1	Average pool, 1000-d FC, Softmax
FLOPs		3.8 × 10^9^

**Table 3 sensors-21-08007-t003:** FCNs architecture.

Layer	Kernel Size. Feature Map	Stride	Output Shape
Input Image	-	-	256 × 256 × 1
Convolution Layer 1	28 × 28 × 256	2	3 × 3
Pooling Layer 1	2 × 2		14 × 14 × 256
Convolution Layer 2	14 × 14× 256	2	3 × 3
Pooling layer 1	2 × 2		7 × 7 × 256
Convolution Layer 3	7 × 7 × 7 × 256 × 7 × 256	2	-
Deconvolution	2 × 2		14 × 14 × 256
Convolution Layer 4	14 × 14 × 256	2	3 × 3
Deconvolution	2 × 2		28 × 28 × 256
Convolution Layer 5	28 × 28 × 256		3 × 3
Convolution	28 × 28 × C		28 × 28 × 1
Output Layer	-	-	1

**Table 4 sensors-21-08007-t004:** The performance of fetal heart standard view.

CNNs Architecture	Performance (%)			
	View	mAP	IoU	DCS
ResNet 50	3VT	96.59	81.76	90.58
	4CH		87.17	90.93
	LVOT		66.29	86.55
	RVOT		84.64	90.73
ResNet 101	3VT	91.85	84.85	81.76
	4CH		79.63	87.17
	LVOT		46.80	66.29
	RVOT		83.55	84.64
Mobilenetv1	3VT	94.87	79.69	89.88
	4CH		87.40	82.35
	LVOT		68.59	80.78
	RVOT		79.31	86.42

**Table 5 sensors-21-08007-t005:** The IoU and the DCS performance for heart defect segmentation.

Position of Heart Defect	Intra-Patient	Inter-Patient	Intra-Patient	Inter-Patient
	IoU (%)		DCS (%)	
Hole in atria	62.72	55.99	77.74	67.69
Hole in ventricle	54.83	42.07	68.26	48.89
Hole in atria and ventricle	58.36	40.54	60.20	52.63

**Table 6 sensors-21-08007-t006:** The mAP performance for heart defect detection.

Intra-Patient	Inter-Patient
98.30%	82.42%

**Table 7 sensors-21-08007-t007:** Research benchmarks with other medical object detection with CNNs techniques.

Author	Method	Object	mAP
Rezvy et al. [17]	Modified Mask-RCNN	Endoscopy disease	0.51
Lin et al. [19]	CNNs-based YOLACT	Psoriasis skin	0.85
Shenavarmasouleh et al. [25]	Mask-RCNN	Exudates and microaneurysms	0.43
Pai et al. [26]	VGG16 and Mask-RCNN	Brain tumor	0.90
Masood et al.	Mask-RCNN	Brain tumor	0.94
Cai et al. [28]	Mask-RCNN	Pulmonary nodule	0.88
Chiao et al. [29]	Mask-RCNN	Breast tumor	0.75
Vuola et al. [30]	Mask-RCNN U-Net Ensemble	Nuclei	0.52
Long et al.	Probability-Mask-RCNN	Pulmonary embolism	0.81
Our proposed model	ResNet50 and Mask-RCNN	Fetal heart defect	0.98

## Data Availability

Not Applicable.

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
