# Peer review of "Deep Learning-Based Computer-Aided Fetal Echocardiography: Application to Heart Standard View Segmentation for Congenital Heart Defects Detection"

_sensors, 2021, doi:10.3390/s21238007_

Round 1

Reviewer 1 Report

The authors present a deep learning approach to CHD in a well organized and presented format. Overall the methodology and set up are appropriate.

Some further discussion on possible bias in the data population would add to the robustness of the work. This topic is being addressed more and more where models would eventually generalize to the world population but are trained on subjects from a specific origin in the world and may not generalize as well as expect (such as the issue with skin color and PPG sensors for example). Are there any ideas or concerns about the fetal images? The dataset was reasonable well-balanced between healthy and unhealthy subjects, but would the sex or ethnic background of the subject data population have any influence on the ability of the model to generalize?

Reviewer 2 Report

The paper proposes a methodology for deep learning-based computer-aided fetal heart echocardiography examinations with an instance segmentation approach. The paper is well prepared and well presented. The methodology is sound, and the experimental results show the viability of the proposed approach. I have only a few minor comments:

  1. The dataset is imbalanced. How did you deal with imbalance? Did you do balancing to improve the result?
  2. Explain the semantics of color markings and abbreviations used in images in Figures 2-5, 10-12, 14 (in the caption of each figure).
  3. The math notation used in Equation 3 is strange.
  4. Figures 8 and 9: the use of cumulative bar plots makes it difficult to compare the values. Consider using some alternative plot such as spider plot.
  5. Lines 505-506: the logic behind the statement how you avoided overfitting is not clear. Specifically, what is the training stoppage criterion? How did it help to avoid/prevent overfitting?
  6. Table 7: authors used different datasets, so the results are not comparable.
  7. Discuss the limitations of the proposed methodology and threats-to-validity of the experimental results.
  8. Improve the conclusions. Use the main numerical findings from the experiments to support your claims.
